# Negative-Pressure Ventilation in Neuromuscular Diseases in the Acute Setting

**DOI:** 10.3390/jcm11092589

**Published:** 2022-05-06

**Authors:** Anna Annunziata, Cecilia Calabrese, Francesca Simioli, Antonietta Coppola, Martina Flora, Antonella Marotta, Valentina Di Spirito, Francesco Didonna, Marcellino Cicalese, Giuseppe Fiorentino

**Affiliations:** 1Department of Intensive Care, AORN Ospedali dei Colli, Via Leonardo Bianchi, 80131 Naples, Italy; francesimioli@gmail.com (F.S.); antonietta.coppola84@gmail.com (A.C.); martina.floraa@gmail.com (M.F.); austen.anto@gmail.com (A.M.); valentina.dispirito@ospedalideicolli.it (V.D.S.); didonnaf@gmail.com (F.D.); giuseppefiorentino1@gmail.com (G.F.); 2Department of Translational Medical Sciences, University of Campania ‘Luigi Vanvitelli’, AORN Ospedali dei Colli, Via Leonardo Bianchi, 80131 Naples, Italy; cecilia.calabrese@unicampania.it; 3Department of Surgery, Unit of Thoracic Surgery, AORN Ospedali dei Colli, Via Leonardo Bianchi, 80131 Naples, Italy; marcellinocicalese@libero.it

**Keywords:** neuromuscular diseases, negative-pressure ventilation, iron lung, poncho, cuirass

## Abstract

Mechanical ventilation started with negative-pressure ventilation (NPV) during the 1950s to assist patients with respiratory failure, secondary to poliomyelitis. Over the years, technological evolution has allowed for the development of more comfortable devices, leading to an increased interest in NPV. The patients affected by neuromuscular diseases (NMD) with chronic and acute respiratory failure (ARF) may benefit from NPV. The knowledge of the available respiratory-support techniques, indications, contraindications, and adverse effects is necessary to offer the patient a personalized treatment that considers the pathology’s complexity.

## 1. Introduction

Patients with neuromuscular diseases (NMD) can present acute and chronic respiratory insufficiency throughout their life and require respiratory support. Although most NMD share common clinical and functional characteristics, they need to be differentiated because some show peculiarities that determine evolution and potential complications. NMD are rare; however, the large number of pathologies determines a great number of individuals who need respiratory assistance. In particular, Duchenne muscular dystrophy, myotonic dystrophy, spinal muscular atrophy, Pompe disease, and amyotrophic lateral sclerosis lead to respiratory muscle weakness. Typically, this results in an ineffective cough, clearance impairment, airway obstruction, restrictive lung disease, swallowing disorders with dysphagia and aspiration, recurrent respiratory infections, sleep-disordered breathing, alveolar hypoventilation, and respiratory failure. Pulmonary complications are the most frequent cause of death. Sialorrhea is also a common and problematic symptom of patients affected by neuromuscular diseases associated with bulbar and/or facial-muscle dysfunction. Thicker and tenacious oral and pharyngeal secretions may result from the drying management of sialorrhea. Ineffective cough and gastroesophageal reflux can lead to aspiration pneumonia. In the beginning, such patients may be asymptomatic but require a regular and careful clinical follow-up because they may develop severe respiratory failure and pneumonia. Sometimes, patients need resuscitation maneuvers due to swallowing disorders.

NMD patients show a restrictive ventilatory syndrome. In addition, the unfavorable conformation of the rib cage can determine areas of atelectasis that—in the context of acute lung injury—can worsen the ventilation/perfusion imbalance, leading to hypoxemia that often requires noninvasive respiratory support.

Negative-pressure ventilation (NPV) was used on a large scale in the 1950s to treat patients with respiratory failure postpoliomyelitis. The first devices were very bulky, not transportable, and interfered with patient care and hygiene. Over the years, positive-pressure ventilation has become the treatment of choice for acute and chronic respiratory failure in neuromuscular diseases; however, technological innovations have allowed a new use of NPV in acute and chronic situations that offer an additional option of treatment in patients sometimes dependent even 24 h a day 24 h/day on respiratory support.

## 2. NPV Interfaces 

The interfaces for NPV are as follows: Section 2.1. Iron lung, Section 2.2. Poncho, Section 2.3. Cuirass. 

### 2.1. Iron Lung

The name “iron lung” comes from the first ventilator designed by Philip Drinker and Louis Agassiz Shaw, a metallic machine. It consisted in a cylindrical ventilation chamber in which the patient lied inside except for his/her head [1,2]. More recently, the metallic structure of the chamber was replaced by lighter materials (fiberglass, aluminum, polycarbonate, and various plastics) in order to improve both the weight and transport. In some models (Porta-Lung), the ventilator pump is separated from the cylindrical chamber, which is available in different sizes.

The iron lung creates a negative pressure inside the chamber during the inspiration phase, which guarantees the uplift of chest and abdomen, and therefore a downward movement of the diaphragm of the patient lying in a supine position. The uplift of the chest and the downward movement of the diaphragm result in a volumetric increase in the chest and in a consequent inlet of air. Exhalation is allowed from the elastic return of the thoracic-pulmonary system, or eventually by the application of a modifiable positive pressure. The pressure delivery is provided by a high-power pump/blower governed by a digital electronic control unit. The machine can operate in both assisted and controlled mode (sometimes only in controlled mode). Through some portholes located on the chamber, it is possible to access and manipulate the patient’s body, position different probes and catheters, or take blood samples. Since the patient’s head is positioned outside the chamber, the patient can talk, eat, drink, sneeze, cough, and expectorate; in addition, it is possible to perform oral-pharyngeal aspirations as well as fibrobronchoscopy and apply devices for oxygen therapy. The iron lung is equipped by alarms and also by a heating system to improve the patient’s comfort and compliance. 

### 2.2. Poncho

The poncho, also called “wrap” or “pneumo-wrap”, is a ventilatory interface that works synergically with the NPV ventilator to generate an effective negative extrathoracic pressure. The pressure is delivered inside a poncho-shaped suit, which the patient has to wear, closed at the neck, wrists, and ankles with straps that ensure airtightness. The empty space between the suit and the patient’s body is guaranteed by a metal/plastic net connected to a supporting plane placed at the back. The poncho is often made with GORE-TEX, and is available in different sizes. The ventilator connected to the poncho can operate both in controlled and assisted mode with sensors positioned at the level of the upper airways.

### 2.3. Cuirass

The cuirass (Figure 1) is a ventilatory interface that works with an NPV ventilator. This interface is shaped as a cuirass or shell made with plastic material. It is positioned on the patient’s chest and held with fabric straps [3]. Many sizes and shapes of the shells are available that allow to accommodate even babies. Some are customized to perfectly fit the patient’s body. The negative pressure is applied inside the shell, while the edges have to be skintight in order to avoid air leaks and generate the pressure variations on the chest. 

Similarly to the “poncho”, the cuirass connected to the ventilator can operate in both controlled and assisted modes. The cuirass applies the negative pressure on a lower surface, but compared to the poncho, it can assist the patient during the exhalation phase by providing a modifiable positive pressure. 

## 3. NPV Mode

At present, there are four different ways to provide NPV: (1) cyclical negative pressure; (2) negative and positive pressure; (3) continuous negative extrathoracic pressure (CNEP); (4) negative pressure/CNEP [4].

(1) Cyclical negative pressure: the negative pressure is delivered during inspiration, while expiration is passive.

(2) Negative/positive pressure: a subatmospheric pressure is delivered during inspiration and a positive pressure during expiration. This biphasic mode can be applied by cuirass and has shown to enhance tidal volume in patients with chest-wall deformities in comparison to the only NPV [5]. 

(3) Continuous negative extrathoracic pressure (CNEP): the ventilator generates a constant negative pressure during the whole respiratory cycle while the patient spontaneously breathes. In animal models, CNEP and positive end-expiratory pressure (PEEP) have shown similar physiological effects [6].

(4) Negative pressure/CNEP: the ventilator has a CNEP mode but there are some inspiratory cycles in which the negative pressure is increased. 

## 4. NPV Candidacy

The first use of NPV dates back to the polio epidemic. These patients, in which the neuromuscular impairment was secondary to the neurological damage induced by the poliovirus, required both an acute and chronic respiratory support [4]. Patients affected by neuromuscular diseases with respiratory failure have many clinical and physiological frailties, and when the respiratory support is required, it is very important to offer different types of interfaces and different pulmonary-ventilation options. Patients requiring 24 h a day a respiratory support can benefit from NPV in alternation with positive-pressure ventilation.

The candidates to NPV are usually patients who cannot adapt to noninvasive ventilation (NIV) because they do not tolerate a facial mask due to facial deformity, are claustrophobic, or produce excessive airway secretions. Considering the potential effects on cardiopulmonary circulation, NPV is also beneficial in young patients and children who have undergone complex cardiac reconstructive surgery.

There are different indications and contraindications for the use of NPV that also depend on the disease. In neuromuscular patients with respiratory failure, they are listed in Table 1 [7]. Obstructive sleep apnea syndrome and severe obesity can be contraindications, due to the possible collapse of the upper respiratory tract and because the available sizes are limited. The interfaces may not be wearable due to severe kyphoscoliosis or recent abdominal surgery. For the mechanism of action, the negative pressure that determines thoracic traction is not usable in the case of thoracic fractures.

Over the years, the technological evolution has allowed lighter and more comfortable devices starting from the iron lung to the cuirass, up to the soft cuirass available today. Despite this technological progress, to date, no guidelines have been set on the indication and modality of NPV.

## 5. NPV in Acute Settings in Neuromuscular Diseases

NPV was used for the first time to treat the acute respiratory failure (ARF) of neuromuscular patients during the poliomyelitis epidemic [4]. In that period, several studies demonstrated a reduction of approximately 50% of mortality of spinal polio patients [8,9,10,11]. Later, few uncontrolled studies investigated the effect of NPV in the treatment of neuromuscular patients with ARF in the setting of the respiratory intensive care unit (ICU) (Table 2). Libby et al. demonstrated the efficacy of the iron lung in avoiding the need for endotracheal intubation of 20 patients affected by ARF due to severe kyphoscoliosis, which also included one post-poliomyelitis patient [12]. Braun et al. studied three patients with a diagnosis of amyotrophic lateral sclerosis and two with Duchenne muscular dystrophy in ARF; NPV, provided by pneumowrap, successfully managed the respiratory failure in one patient and allowed the weaning from invasive mechanical ventilation in two other patients [13]. Garay et al. studied three patients with a diagnosis of kyphoscoliosis, two post-poliomyelitis, one with muscular dystrophy, and two in coma from carbon-dioxide narcosis treated with NPV; all patients were discharged and continued NPV treatment at home together with mouthpiece positive-pressure ventilation performed during daytime hours [14]. NPV can also be an option in acute settings for weaning from oro-tracheal intubation or for tracheostomized patients. There is some evidence regarding successful weaning from intermittent positive-pressure ventilation via endotracheal intubation by means of an NPV [15]. In a retrospective study, Corrado et al. demonstrated the effects of NPV for the treatment of ARF of 15 neuromuscular patients, of which 7 affected by amyotrophic lateral sclerosis, 5 by muscular dystrophy, 2 by myasthenia gravis, and 1 by multiple sclerosis [16]. Upon admission in the respiratory ICU, all patients presented severe hypoxemia and hypercapnia with respiratory acidosis. NPV was performed with success to treat ARF in 12 out of 15 patients (80%). Recently, NPV was successful in a case of nemaline myopathy complicated by pneumothorax; the positive airway pressure was avoided because it could have aggravated the pneumothorax, while NPV provided an adequate respiratory support moving the patient’s chest wall with the negative pressure [17]. Finally, NPV was used, in combination with other respiratory supports, to allow weaning from orotracheal intubation in a patient affected by ARF with hypoxemia, hypercapnia, and respiratory acidosis due to girdle dystrophy; NPV was continued at home [18]. Some authors described prolonged use of intermittent negative pressure in a group of patients affected by Duchenne muscular dystrophy. The patients preferred negative-pressure ventilators in comparison to the positive-pressure ones because, in their opinion, they were effective, comfortable, and left the upper airway free [19]. 

There are few data in the literature regarding the use of NPV in the pediatric intensive setting. The treatment of ARF in a pediatric ICU is a challenge due to the risk of invasive positive-pressure ventilation in terms of barotrauma, volutrauma, and atelectrauma. Noninvasive ventilation techniques can avoid these problems and NPV can represent an alternative strategy to noninvasive positive-pressure ventilation in pediatric patients with abnormal facial morphologies, anxiety, excessive oropharyngeal secretions and emesis [20,21]. In literature, there are few data regarding neuromuscular pediatric patients affected by ARF treated with NPV. In a retrospective study enrolling 15 patients with neuromuscular disorders, Hassinger et al. observed that 70% of patients responded to NPV [21]. Moreover, a recent retrospective study showed a response to NPV in the 69% of a cohort of pediatric patients with ARF of various etiology: among them, three patients were affected by neuromuscular diseases, and two of them showed good results [22]. 

## 6. Advantages and Limitations of NPV in Acute Setting

The major advantage of NPV is the preservation of physiological functions, such as speech, cough, swallowing, and feeding. These aspects are very relevant for patients who are continuously ventilated, who may be dependent on respiratory support before or after the acute event. Moreover, NPV, unlike positive-pressure ventilation (facial or total-face interfaces are generally used in acute settings), allows fibrobronchoscopy to be performed for various procedures without interruption from the respiratory support.

Limits of NPV have to be considered: firstly, the lack of protection of the upper airways, and secondly, upper-airway obstruction due to both the downfall of the tongue on the posterior pharyngeal wall and the lack of preinspiratory upper-airway muscle activation. These problems are particularly relevant in comatose patients, in those with neurological disorders associated with bulbar dysfunction (e.g., bulbar amyotrophic lateral sclerosis), and in patients affected by apnea during sleep. In unconscious patients with normal bulbar function, the placement of a nasogastric tube and the positioning of an oropharyngeal airway can reduce the problem of aspiration pneumonia and upper-airway closure due to the collapse of the tongue.

The intermittent upper-airway collapse leading to obstructive apnea is a potential risk associated to NPV in patients affected by neuromuscular diseases. This complication was firstly recognized several decades ago: the glottic closure was observed in normal subjects hyperventilated in iron lungs and was thought to represent a protective reflex against hyperventilation. The lack of pre-inspiratory activation of upper-airway muscles predisposes these structures to collapse when exposed to negative pressure. Hill et al. observed some clustering of disordered breathing events during REM sleep in Duchenne muscular dystrophy patients during NPV, when the breathing pattern was irregular and the upper-airway tone was reduced [23]. For this reason, patients should be carefully monitored and a mixed ventilation with NPV and nocturnal Continuous Positive Airway Pressure (CPAP) alone or during NPV can be an option [24,25]. The possibility of combining NPV with CPAP has been reported to be successful in overcoming upper-airway obstruction following extubation in infants weaned from invasive ventilation by means of NPV [20].

The most common side effects reported with NPV are poor patient compliance, upper-airway obstruction, and musculoskeletal pain [16] (Table 3). These are side effects that can arise with the different types of respiratory support and that must necessarily be taken into consideration for each individual, in the patient treated in an acute setting and especially in dependent patients 24 h a day.

In case of prolonged use, as happens with all interfaces, the appearance of decubitus is possible. 

## 7. Conclusions

All noninvasive ventilatory techniques, reducing the need for endotracheal intubation and its related complications and difficulty in weaning, should be considered potentially advantageous in certain subgroups of patients. Although the literature suggests the effectiveness of NPV in the treatment of ARF in patients with neuromuscular diseases, many important aspects still need to be clarified. The lack of expertise in many centers probably limits its use. Prospective controlled studies have to be performed in order to evaluate the effect of NPV on both mortality and length of hospital stay in patients affected by neuromuscular disorders who develop ARF.

NPV should also be considered as an option for the treatment of neuromuscular patients with ARF at home. In addition, NPV can be utilized by patients with chronic respiratory failure, who, for a variety of reasons, are unable to tolerate noninvasive positive-pressure ventilation or are dependent on mechanical ventilation 24 h a day; in this case the possibility of alternating different devices can prevent the onset of complications and decubitus. Although some side effects of NPV can occur in neuromuscular patients, the overall benefits in this group of patients outweigh its side effects. 

## Figures and Tables

**Figure 1 jcm-11-02589-f001:**
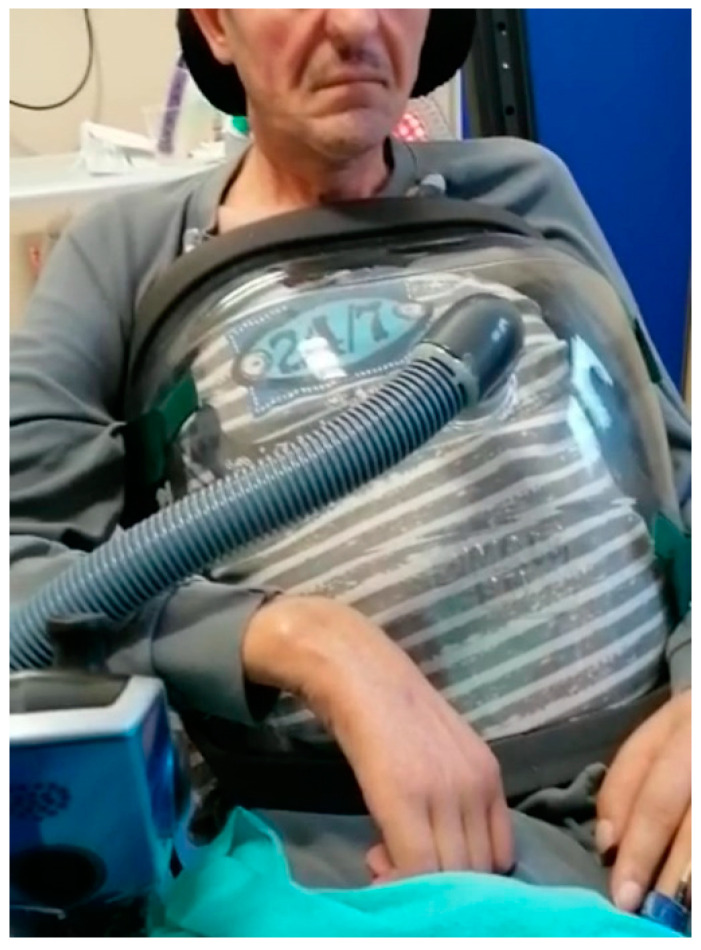
A patient with ALS during NPV.

**Table 1 jcm-11-02589-t001:** NPV indications and contraindications.

Indications	Contraindications
Severe facial decubitus	Sleep apnea syndrome
Interface intolerance	Severe obesity
Facial deformity	Severe kyphoscoliosis
Severe respiratory acidosis	Rib fractures
Severe hypercapnic encephalopathy	Recent abdominal surgery

**Table 2 jcm-11-02589-t002:** Studies on NPV in NMD patients with ARF. AMC: arthrogryposis multiplex congenital, EI: endotracheal intubation, ARF: acute respiratory failure, ALS: amyotrophic lateral sclerosis, DMD: Duchenne muscular dystrophy, PPV positive-pressure ventilation, NPPV: noninvasive positive-pressure ventilation MD: muscular dystrophy, MG: myasthenia gravis, MS: multiple sclerosis, LGMDs: limb–girdle muscular dystrophy, CNEP: continuous negative extrathoracic pressure.

Reference	Study Design	Sample Size (N°)	Age Mean	Neuromuscular Diseases	NPV	Findings
Libby et al. 1982 [12]	Case series	20	52 years (13–78)	Severe scoliosis or kyphosis with ARF (6 poliomyelitis, 1 AMC)	Iron lung	The only poliomyelitis patient treated with NPV avoided EI
Braun et al. 1987 [13]	Case series	5	49 years (19–70)	3 ALS, 2 DMD	Intermittent Pneumowrap	1 ALS patient avoided EI, 3 ALS patients slowed pulmonary deterioration, 2 DMD patients were weaned from PPV
Garay et al. 1981 [14]	Case series	8	(33–71)	Alveolar hypoventilation (3 kyphoscoliosis, 2 postpoliomyelitis, 1 MD, 1 1° alveolar hypoventilation, 1 pneumonectomy with phrenic nerve crush)	iron lung (nocturnal NPV + daytime MPV)	All patients discharged and home treatment during 10 years
Corrado et al. 1995 [16]	Retrospective study	15	-	7 ALS, 5 MD, 2 MG, 1 MS	Iron lung	Successful treatment in 12 patients
Hino et al. 2016 [17]	Case Report	1	11 years	Nemaline myopathy complicated by tension pneumothorax undergoing NIPPV	Biphasic cuirass ventilation	Successful treatment, air leak resolution
Imitazione et al. 2021 [18]	Case Report	1	56 years	Limb–girdle muscular dystrophy (LGMDs)	Poncho NPV + HFNC	Successful treatment in hospital and at home after refusing NIV.
Hassinger et al. 2017 [21]	Retrospective chart review	233	15.5 months (7.6–39.6)	15 neuromuscular diseases	Cuirass (CNEP or biphasic)	8 patient responders
Nunez et al. 2019 [22]	Retrospective cohort study	118	12 months	3 neuromuscular diseases	Biphasic cuirass ventilation	2 patient responders 1 nonresponder

**Table 3 jcm-11-02589-t003:** Side effects of NPV.

Side Effects
Apneas Intolerance of corset Severe decubitus Lack of protection of the upper airway Musculoskeletal pain

## Data Availability

Not applicable.

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
