# Peer review of "Negative-Pressure Ventilation in Neuromuscular Diseases in the Acute Setting"

_jcm, 2022, doi:10.3390/jcm11092589_

Round 1
Reviewer 1 Report
The Title refers to the use of negative pressure ventilation in the acute setting, but most of the paper deals with the chronic use of NPV.
The Abstract does not reflect the paper very well. More than half of the Abstract is about the history of negative pressure ventilation (NPV). It is not quite correct to say that all patients with respiratory failure associated with neuromuscular diseases will benefit from NPV as most of these patients will now be managed with non-invasive positive pressure ventilation (PPV).
The Introduction gives a good description of the different modes and interfaces of NPV. However, it might be more useful if it also included some background about respiratory failure in neuromuscular diseases and how non-invasive ventilation may help. Some explanation about the role of nocturnal ventilation in some patients and the need for twenty-four ventilation in others would also help to orientate the reader. As PPV is hardly mentioned, the paper creates the impression that NPV is the standard of care in these patients, whereas most patients needing non-invasive ventilation will be on PPV. This needs an explanation in the Introduction, and throughout the paper it needs to be made much clearer in which patients NPV would be suitable rather than PPV, which seems to have been overlooked..
The opening sentence in Section 2 needs correction as many patients with neuromuscular diseases do not need NIV. The statement that animal models have shown superiority of NPV over PPV needs references. The papers supporting NPV in people are quite old and predate the development and widespread use of PPV. To be able to justify the use of NPV over PPV, you need to present data from studies with a direct comparison to support this assertion in 2022.
In Section 3 the choice in clinical practice is not really between intubation and NPV, but rather between PPV and NPV to hopefully avoid intubation. This raises the issue of making a distinction between type 1 and type 2 respiratory failure and the limited role, if any, of NPV or PPV in patients with acute type 1 respiratory failure.
The issues raised in Tables 1 and 3 need fuller explanations in the text.
Author Response
The Title refers to the use of negative pressure ventilation in the acute setting, but most of the paper deals with the chronic use of NPV.
A: In accordance with this consideration, we have eliminated the brief introduction and the paragraphs relating to chronic neuromuscular pathology and tried to highlight the literature data relating to the case series of patients treated in ICU or in acute on chronic (ARF, pneumothorax, extubation )
The Abstract does not reflect the paper very well. More than half of the Abstract is about the history of negative pressure ventilation (NPV). It is not quite correct to say that all patients with respiratory failure associated with neuromuscular diseases will benefit from NPV as most of these patients will now be managed with non-invasive positive pressure ventilation (PPV).
A:Thanks for the comment, I have modified in accordance with the suggestion. If necessary, i will make further changes.
The Introduction gives a good description of the different modes and interfaces of NPV. However, it might be more useful if it also included some background about respiratory failure in neuromuscular diseases and how non-invasive ventilation may help. Some explanation about the role of nocturnal ventilation in some patients and the need for twenty-four ventilation in others would also help to orientate the reader. As PPV is hardly mentioned, the paper creates the impression that NPV is the standard of care in these patients, whereas most patients needing non-invasive ventilation will be on PPV. This needs an explanation in the Introduction, and throughout the paper it needs to be made much clearer in which patients NPV would be suitable rather than PPV, which seems to have been overlooked..
A: thanks for review, In accordance with what was suggested by both reviewers, we have modified the paper, developing a brief introduction; we have removed some parts from the body of the text (too introductory - as suggested by rev 2). We made it clear in the introduction that NPV is an interesting alternative option, which we need to know for the best possible treatment for our patient, but that the gold standard is PPV.
The opening sentence in Section 2 needs correction as many patients with neuromuscular diseases do not need NIV. The statement that animal models have shown superiority of NPV over PPV needs references. The papers supporting NPV in people are quite old and predate the development and widespread use of PPV. To be able to justify the use of NPV over PPV, you need to present data from studies with a direct comparison to support this assertion in 2022.
A: We have changed this point. In fact, after your kind review, we want to clarify that the purpose of the paper is not to propose NPV as an absolute alternative to PPV, but as a possible alternative. Particularly for those patients with neuromuscular pathology who are dependent on NIV and who become complicated with ARF.
We have included in the table the works in the literature relating to patients treated in an acute setting.
In Section 3 the choice in clinical practice is not really between intubation and NPV, but rather between PPV and NPV to hopefully avoid intubation. This raises the issue of making a distinction between type 1 and type 2 respiratory failure and the limited role, if any, of NPV or PPV in patients with acute type 1 respiratory failure.
A: Thanks for the comment. We have tried to focus attention on actual practical advantages and limitations, avoiding confusion in the reader and reducing the emphasis that probably emerged from the previous version.
In our opinion and experience, it is useful to know the possibility of using a respiratory support technique such as NPV, as an additional weapon and not a substitute for positive pressure ventilation. For some patients, NPV may represent the best technique or an additional technique, to be used in alternation with positive pressure.
We hope that the changes made in accordance with the suggestions proposed by the reviewers in the paper will be clearer than previous manuscript.
The issues raised in Tables 1 and 3 need fuller explanations in the text.
A: Thanks for the suggestion. We have included others clarifications on the tables in the text
Thank you for the attention and for your time
Reviewer 2 Report
General Comments/Questions:
It would be useful to have a photograph or illustration of each of the three types of NPV devices.
I think that this manuscript would benefit from more headings to help with transitions from topic to topic. I have noted my suggestions in the specific suggestions below.
Is NPV as effective as positive pressure ventilation (invasive or noninvasive) for those patients with severe paresis/complete paralysis? For example, could a patient with completely paralyzed chest wall muscles be ventilated adequately with a chest cuirass as they are with a positive pressure ventilator?
Specific Comments/Questions:
p. 1: I would change the major heading of “1. Introduction” to “1. NPV Interfaces”
p. 1, line 3 under Iron Lung: Change “lays” to “lies”
p. 2, line 4 under Poncho: Change “close” to “closed”
p. 2, line 7: I believe goretex is capitalized and trademarked…? GORE-TEX
p. 2, line beginning with “At present, there are four different ways to provide NPV:…”: I think this should be a new major section (same level as the current “Introduction”) called something like “2. NPV Delivery Options”
p. 2, beginning with the sentence “The first use of NPV dates back to….”: I would introduce this with another major heading, for example: “3. NPV Candidacy”
p. 3, under the heading “NPV in acute setting…”: The first two paragraphs do not seem necessary to me for the purpose of this manuscript. I would start this section with the paragraph “NPV was used for the first time to treat the acute respiratory failure…”
p. 4, line 4: I suggest beginning a new paragraph at “NPV can also be an option in ….”
p. 4, line 12: Does the word “successful” in this context mean “successfully weaned from the ventilator”?
p. 4: The paragraph beginning with “The treatment of ARF in a pediatric ICU…” seem disjointed and not very informative. Perhaps this paragraph could be deleted…?
p. 4: The last two paragraphs (one starting with “Although these studies…” and one starting with “NPV should be considered…” do not seem to fit here. Perhaps they go in a “conclusion” section.
p. 5, line 3 under “Advantages and clinical…”: I don’t believe NIV has been defined. Is this referring to noninvasive ventilation with a face/nose interface?
p. 6, lines 18-19: What is meant by “compliance” in this context”? Patient compliance to use the device? Chest wall or airway compliance?
Author Response
It would be useful to have a photograph or illustration of each of the three types of NPV devices.
We have photos only of the third interface, the curiass. We added it.
I think that this manuscript would benefit from more headings to help with transitions from topic to topic. I have noted my suggestions in the specific suggestions below.
Thanks for the review
Is NPV as effective as positive pressure ventilation (invasive or noninvasive) for those patients with severe paresis/complete paralysis? For example, could a patient with completely paralyzed chest wall muscles be ventilated adequately with a chest cuirass as they are with a positive pressure ventilator?
Negative non invasive mechanical ventilation offers a modality of respiratory support analogous to positive pressure. It can also be used with completely paralyzed patients (the first large-scale use was during the polio era). Our experience is mainly in patients with neuromuscular disease, ALS, duchenne, dystrophies, secondary tetraparesis, respiratory failure in patients with advanced cachexia.
In addition to the aspects described in the paper, the problems that have limited its use are the size of the equipment and the care and hygiene of the patient. Furthermore, it can be a valid tool for the rotation of aids to avoid decubitus and complications
Specific Comments/Questions:
- 1: I would change the major heading of “1. Introduction” to “1. NPV Interfaces”
ok, thanks
- 1, line 3 under Iron Lung: Change “lays” to “lies”
ok, thanks
- 2, line 4 under Poncho: Change “close” to “closed”
ok, thanks
- 2, line 7: I believe goretex is capitalized and trademarked…? GORE-TEX
ok, thanks
- 2, line beginning with “At present, there are four different ways to provide NPV:…”: I think this should be a new major section (same level as the current “Introduction”) called something like “2. NPV Delivery Options”
ok thanks for the kind suggestion
- 2, beginning with the sentence “The first use of NPV dates back to….”: I would introduce this with another major heading, for example: “3. NPV Candidacy”
ok thanks for the kind suggestion
- 3, under the heading “NPV in acute setting…”: The first two paragraphs do not seem necessary to me for the purpose of this manuscript. I would start this section with the paragraph “NPV was used for the first time to treat the acute respiratory failure…”
ok I have deleted in this paragraph. Furthermore As requested by rev 1 we have reworked a part and structured a brief introduction
- 4, line 4: I suggest beginning a new paragraph at “NPV can also be an option in ….”
ok
- 4, line 12: Does the word “successful” in this context mean “successfully weaned from the ventilator”?
We correct with :NPV was performed with success to treat ARF in 12 out of 15 patients
- 4: The paragraph beginning with “The treatment of ARF in a pediatric ICU…” seem disjointed and not very informative. Perhaps this paragraph could be deleted…?
Ok I modify the point
- 4: The last two paragraphs (one starting with “Although these studies…” and one starting with “NPV should be considered…” do not seem to fit here. Perhaps they go in a “conclusion” section.
ok thanks for the kind suggestion
- 5, line 3 under “Advantages and clinical…”: I don’t believe NIV has been defined. Is this referring to noninvasive ventilation with a face/nose interface?
We added: in acute setting commonly face or total face interface
- 6, lines 18-19: What is meant by “compliance” in this context”? Patient compliance to use the device? Chest wall or airway compliance?
Ok I modify “patient compliance”
Thank you for your attention and your time
Round 2
Reviewer 2 Report
Excellent revision! I have just a few very minor suggested edits:
p. 4, paragraph 2 under NPV Candidacy: Change “24/24” to “24/7” (if you mean “all the time” or 24 hours/day, 7 days/week) or it should read “24/24 hours”
p. 4, heading “NPV in acute setting in …”: Should it be “settings”?
p. 7, line 3 under heading “6.”: It should read “… preservation of physiological functions, such as…”
p. 7, line 6 under heading “6.”: It should read something like “… (in acute setting wherein facial interfaces are commonly used)”
